# *Caenorhabditis elegans* nuclear RNAi factor SET-32 deposits the transgenerational histone modification, H3K23me3

Lianna Schwartz-Orbach[1], Chenzhen Zhang[1], Simone Sidoli[2], Richa Amin[1], Diljeet Kaur[1], Anna Zhebrun[1], Julie Ni[1], Sam G Gu[1]*

[1]Department of Molecular Biology and Biochemistry, Rutgers the State University of New Jersey, Piscataway, United States; [2]Department of Biochemistry, Albert Einstein College of Medicine, Bronx, New York, United States

**Abstract** Nuclear RNAi provides a highly tractable system to study RNA-mediated chromatin changes and epigenetic inheritance. Recent studies have indicated that the regulation and function of nuclear RNAi-mediated heterochromatin are highly complex. Our knowledge of histone modifications and the corresponding histonemodifying enzymes involved in the system remains limited. In this study, we show that the heterochromatin mark, H3K23me3, is induced by nuclear RNAi at both exogenous and endogenous targets in *C. elegans*. In addition, dsRNA-induced H3K23me3 can persist for multiple generations after the dsRNA exposure has stopped. We demonstrate that the histone methyltransferase SET-32, methylates H3K23 *in vitro*. Both *set-32* and the germline nuclear RNAi Argonaute, *hrde-1*, are required for nuclear RNAi-induced H3K23me3 *in vivo*. Our data poise H3K23me3 as an additional chromatin modification in the nuclear RNAi pathway and provides the field with a new target for uncovering the role of heterochromatin in transgenerational epigenetic silencing.

*For correspondence:
sam.gu@rutgers.edu

**Competing interests:** The authors declare that no competing interests exist.

## Introduction

Nuclear RNAi is an evolutionarily conserved pathway in which small RNAs mediate transcriptional silencing and heterochromatin formation (*Wassenegger, 2000*; *Sienski et al., 2012*; *Pezic et al., 2014*; *Rechavi et al., 2014*; *Martienssen and Moazed, 2015*). Nuclear RNAi plays an important role in genome stability and germline development. It is a highly tractable system for the study of RNA-mediated chromatin regulation and epigenetic inheritance.

*C. elegans* provides a number of unique advantages for the study of nuclear RNAi (*Weiser and Kim, 2019*). Genetic screens have identified numerous protein factors involved in this pathway. In the *C. elegans* germline, nuclear RNAi relies on the Argonaute protein HRDE-1 (*Ashe et al., 2012*; *Buckley et al., 2012*; *Shirayama et al., 2012*). In current models, HRDE-1 binds secondary siRNAs and recruits nuclear RNAi factors, including chromatin modifying enzymes and remodeling factors, to genomic sites of RNAi. We and others have characterized over 150 genomic loci in *C. elegans* that are de-silenced and/or lose repressive chromatin modifications in nuclear RNAi-deficient mutants, the so-called 'endogenous targets' (*Ni et al., 2014*; *McMurchy et al., 2017*). In addition, nuclear RNAi-mediated silencing can be experimentally triggered at actively transcribed genes by exogenous dsRNA administration or piRNA ('exogenous targets') (*Ahringer, 2006*; *Vastenhouw et al., 2006*; *Guang et al., 2010*; *Ashe et al., 2012*; *Gu et al., 2012*; *Shirayama et al., 2012*). Silencing at the exogenous targets can persist for multiple generations. Germline nuclear RNAi-deficient mutants in *C. elegans* exhibit several phenotypes, including progressive sterility

under heat stress (Mrt phenotype) and large-scale de-silencing and chromatin decompaction at the endogenous targets (*Guang et al., 2010*; *Ashe et al., 2012*; *Buckley et al., 2012*; *Shirayama et al., 2012*; *Weiser et al., 2017*; *Fields and Kennedy, 2019*).

There are two known nuclear RNAi-induced histone modifications in *C. elegans*: trimethylation at lysine 27 and lysine 9 of histone H3 (H3K27me3 and H3K9me3). The best studied nuclear RNAi-induced histone modification is H3K9me3, however the function of this histone modification in nuclear RNAi remains unknown (*Burton et al., 2011*; *Gu et al., 2012*; *Mao et al., 2015*; *Kalinava et al., 2017*). Prior models assumed that the heterochromatin mark was required for transcriptional silencing in nuclear RNAi, but surprisingly, H3K9me3 is not essential for silencing maintenance if HRDE-1 is present (*Kalinava et al., 2017*; *McMurchy et al., 2017*; *Woodhouse et al., 2018*; *Lev et al., 2019*). By uncovering the histone methyltransferases (HMTs) responsible for each histone modification, we can better understand their function in nuclear RNAi. Two HMTs, MET-2 and SET-25, are suggested to function sequentially for H3K9 methylation (*Towbin et al., 2012*). In embryos, MET-2 and SET-25 appear to be the sole contributors of H3K9 methylation (*Towbin et al., 2012*; *Garrigues et al., 2015*). However, in adults, nuclear RNAi-mediated H3K9me3 is dependent on a third additional HMT, SET-32 (*Mao et al., 2015*; *Kalinava et al., 2017*; *Spracklin et al., 2017*). While MET-2, SET-25 and SET-32 are all required for the formation of H3K9me3, they are not functionally equivalent. MET-2, but not SET-25, is required for DNA replication stress survival (*Padeken et al., 2019*; *Yang et al., 2019*). SET-25 and SET-32, but not MET-2, are required for the silencing of a piRNA-targeted reporter gene (*Ashe et al., 2012*). In addition, while MET-2, SET-25, and SET-32 were all dispensable for silencing maintenance in nuclear RNAi, SET-32 and, to a lesser extent, SET-25 are required for silencing establishment (*Kalinava et al., 2018*; *Woodhouse et al., 2018*). Given SET-32's unique role in nuclear RNAi, we hypothesized that its biochemical activity may differ from MET-2 and SET-25.

The chromatin landscape in *C. elegans* is dynamically regulated during both somatic and germline development (*Schaner and Kelly, 2006*; *Sidoli et al., 2016b*). From the embryonic stage to adulthood, the two most prominently methylated lysines of histone H3 are H3K27 and H3K23, while H3K9me is proportionally much lower (*Vandamme et al., 2015*; *Sidoli et al., 2016b*). H3K23me has been suggested as a heterochromatin mark in *C. elegans* (*Vandamme et al., 2015*; *Sidoli et al., 2016b*) and *Tetrahymena* (*Papazyan et al., 2014*) and is involved in DNA damage control (*Papazyan et al., 2014*). In comparison to the two classical heterochromatin marks, H3K9me3 and H3K27me3, H3K23me is poorly studied. Almost all histone lysine methylation is catalyzed by SET-domain containing histone methyltransferases (*Cheng et al., 2005*; *Qian and Zhou, 2006*; *Husmann and Gozani, 2019*). Although different HMTs share core catalytic motifs in the SET domain, they can target different lysine residues with high specificity (*Cheng et al., 2005*). In *Tetrahymena*, the SET-domain containing enzyme, EZL3, is required for H3K23me3 *in vivo* (*Papazyan et al., 2014*). In *C. elegans*, loss of SET-32 causes reduced levels of H3K23me1 and H3K23me2 (H3K23me3 was not tested) (*Woodhouse et al., 2018*). No H3K23 HMT has been biochemically validated at the time of this manuscript publication.

In this paper, we determined that SET-32 is an H3K23 methyltransferase *in vitro*. We show that H3K23me3 can be induced by exogenous dsRNA and persists for four generations after the dsRNA exposure has been stopped. H3K23me3 is broadly enriched in *C. elegans* heterochromatic regions, including the endogenous targets of nuclear RNAi. In addition, H3K23me3 at nuclear RNAi targets is dependent on HRDE-1 and SET-32, and, to a lesser extent, MET-2 and SET-25.

## Results

### SET-32 methylates lysine 23 of histone H3 *in vitro*

To determine the enzymatic activity of SET-32, we performed histone methyltransferase (HMT) assays using recombinant GST-SET-32 and [3H]-labeled S-adenosylmethionine (SAM) (*Figure 1—figure supplement 1*). We first tested SET-32's ability to methylate each of the four core histone proteins, and found that GST-SET-32 methylated free and nucleosomal H3, but not H2A, H2B, or H4 (*Figure 1A*). There are four conserved catalytic motifs in the SET-domain family proteins, all of which are found in SET-32 (*Figure 1—figure supplement 2*; *Cheng et al., 2005*). The highly conserved tyrosine residue at position 448 in motif IV is predicted to be one of the catalytic residues of SET-32.

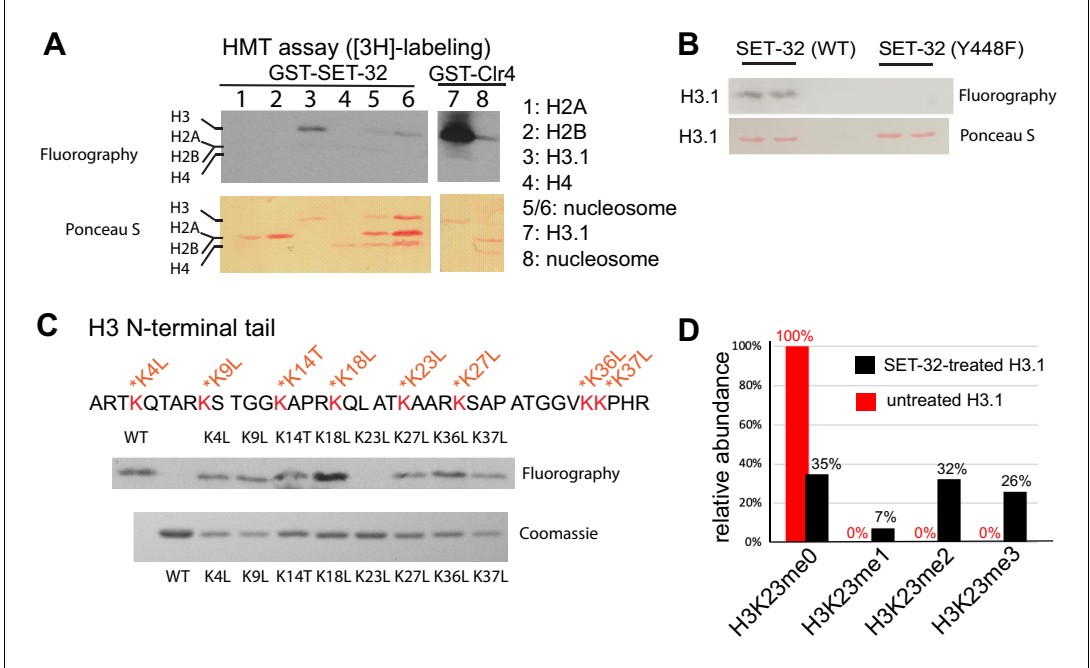

**Figure 1.** SET-32 methylates H3K23 *in vitro*. (**A**) Detecting the HMT activity of GST-SET-32 by [3H]-labeling and fluorography. Individual core histone proteins and *in vitro* assembled mononucleosome made of 601 DNA and recombinant *C. elegans* H2A, H2B, and H3.1, and *Xenopus* H4. *Xenopus* H4 was used because *C. elegans* H4 expression was not successful and there is only one amino acid difference between the two. GST-Clr4 was used as a positive control. (**B**) Fluorography of GST-SET-32 (WT and Y448) HMT assay using histone H3.1. (**C**) Top panel: fluorography of GST-SET-32 HMT assay using WT H3.1 and eight lysine mutants of H3.1. An empty lane was added between the WT H3 and H3K4L for HMT assay to avoid contamination between the WT and H3K4L lanes. Bottom panel: Coomassie staining of WT and mutant H3.1. (**D**) Mass spectrometry analysis of GST-SET-32-treated H3.1 versus untreated H3.1. The percentages of H3K23-containing fragments with H3K23me0, 1, 2, and 3 are indicated above bars.

The online version of this article includes the following figure supplement(s) for figure 1:

**Figure supplement 1.** Recombinant GST-fusion protein purification.
**Figure supplement 2.** Alignment of SET domains of histone methyltransferases.
**Figure supplement 3.** Recombinant histone purification and histone octamer assembly.

In order to determine if this residue is required for SET-32's HMT function, we mutated this tyrosine to phenylalanine (Y448F). The Y448F mutation abolished the HMT activity of GST-SET-32 on H3 (*Figure 1B*).

We next investigated the target lysine of SET-32. The free N-terminal tail domain of H3 contains eight lysine residues available for methylation. To determine whether any of these lysines is required for GST-SET-32's HMT activity, we generated eight mutant H3 proteins. In each mutant, we substituted one of the eight lysines to either leucine or threonine (*Figure 1C*). GST-SET-32 was able to methylate every mutant H3, except for H3K23L (*Figure 1C*), suggesting that the lysine 23 is SET-32's target. We then performed mass spectrometry on GST-SET-32-treated H3, and detected mono, di, and tri-methylation at the K23 position (*Figure 1D*). The untreated control H3 had no H3K23 methylation. We did not detect methylation at any of the other lysine residues of H3. These results agree with published histone mass spectrometry analysis of *set-32* mutant animals, which exhibited reduced H3K23me compared to wild-type animals (*Woodhouse et al., 2018*). Taken together, our data indicate that SET-32 methylates H3K23 *in vitro*.

## Nuclear RNAi triggers transgenerational H3K23me3 at germline genes

Since SET-32 is a nuclear RNAi factor, we hypothesized that H3K23me would be induced by nuclear RNAi. Nuclear RNAi can be triggered by both exogenous dsRNA and mediated by siRNAs at the endogenous targets. We first tested whether H3K23me3 could be induced exogenously, as follows. We chose a well-characterized germline-specific gene, *oma-1*, as the target gene. To induce RNAi in *C. elegans*, we fed worms with *E. coli* expressing *oma-1* dsRNA. Wild type animals were fed with

*oma-1* dsRNA, control dsRNA (GFP), or no dsRNA for 3–4 generations, and the synchronized young adult animals were then collected for ChIP-seq. All antibodies used in this study were validated using western blot and/or immunofluorescence analysis (*Figure 2—figure supplement 1*). In WT animals, *oma-1* dsRNA feeding triggered no detectable change in H3K23me1, a modest increase in H3K23me2, and a dramatic increase in H3K23me3 (*Figure 2A*). The peak of H3K23me3 corresponds to the trigger region of the *oma-1* dsRNA, and spreads approximately 0.5 kb upstream and 1 kb downstream of the *oma-1* gene boundaries. The GFP dsRNA and no dsRNA controls did not show any H3K23 methylation at *oma-1*. In order to compare H3K23me3 with a previously described dsRNA-induced histone mark, we performed side-by-side analysis of H3K23me3 and H3K9me3 and observed closely overlapping profiles (*Figure 2—figure supplement 2*). To verify that H3K23me3 can be induced at other genes, we fed worms with *smg-1* dsRNA and confirmed that H3K23me3 was enriched at the target chromatin in response to *smg-1* dsRNA, with a profile similar to H3K9me3 (*Figure 2B*).

In order to examine the transgenerational dynamics of dsRNA-induced H3K23me3, we performed a heritable RNAi experiment of *oma-1*. WT animals were fed *oma-1* dsRNA for three generations and subsequently moved to dsRNA-free plates and collected for four generations. Similar to H3K9me3, H3K23me3 at *oma-1* persisted for four generations after the RNA-feeding had been stopped (*Figure 2C and D*). These results indicate that RNAi-mediated H3K23me3 is a transgenerational epigenetic effect in *C. elegans*.

## H3K23me3 is a heterochromatic mark in *C. elegans*

In order to further characterize H3K23me in *C. elegans,* we conducted ChIP-seq in WT animals and performed whole-genome analysis. The genomic distribution of H3K23me3 was highly similar to the genomic distribution of H3K9me3 (*Figure 3A–B* and *Figure 3—figure supplements 1* and *2*). We observed H3K23me3 enrichment at constitutive heterochromatin domains: the left and right arms of the five autosomes and the left tip of the X chromosome. Like H3K9me3, we observed the highest peaks of H3K23me3 at the meiotic paring centers (*Figure 3A* and *Figure 3—figure supplement 1*). H3K23me2 and H3K23me1 seem to have a relative uniform distribution in the genome (*Figure 3—figure supplement 1*). In order to quantify our coverage plots, we used boxplot analysis to compare the ChIP-seq signal at the arms of each chromosome compared with the middle (*Figure 3—figure supplement 2*). As expected, H3K9me3 shows statistically significant enrichment at the arms compared to the middle of each chromosome. H3K23me3 also displays enrichment at the arms, however, to a lesser extent. In this analysis, we found that while there were statistically significant differences between the arms and the middle in H3K23me2 and H3K23me1, the magnitude of difference is virtually indiscernible making it highly likely that these differences have little biological significance.

Our results are consistent with previous reports that H3K23me3 is a constitutive heterochromatin mark (*Papazyan et al., 2014*; *Vandamme et al., 2015*; *Sidoli et al., 2016b*). In order to further assess the correspondence of H3K23me3 and H3K9me3, we plotted whole genome coverage of both modifications on a scatter plot and observed a high correlation (Pearson coefficient = 0.8) (*Figure 3B*). Our data confirm previous reports that H3K23me3 is heterochromatic in distribution and correlates highly with H3K9me3.

## H3K23me3 is enriched at endogenous targets of nuclear RNAi

We have previously described a set of endogenous targets of germline nuclear RNAi which consist mainly of LTR retrotransposons and other repetitive genomic elements. These HRDE-1-dependent loci are transcriptionally silenced and enriched for H3K9me3. We examined several of these loci for H3K23 methylation using ChIP-seq (*Figure 3C,E–G*). At the LTR retrotransposon *Cer3*, H3K23me1 signal was double that of the input, however, the signal was still relatively low (*Figure 3C*). We observed a stronger signal for H3K23me2 and a robust signal for H3K23me3 (*Figure 3C*). At the endogenous targets, H3K23me3 closely resembles the H3K9me3 signal (*Figure 3C,E–G*). By contrast, the actively transcribed *cdc-42* locus did not show any enrichment of either H3K9me3 or H3K23me3 (*Figure 3H*). These data demonstrate that H3K23me3 is enriched at endogenous nuclear RNAi targets.

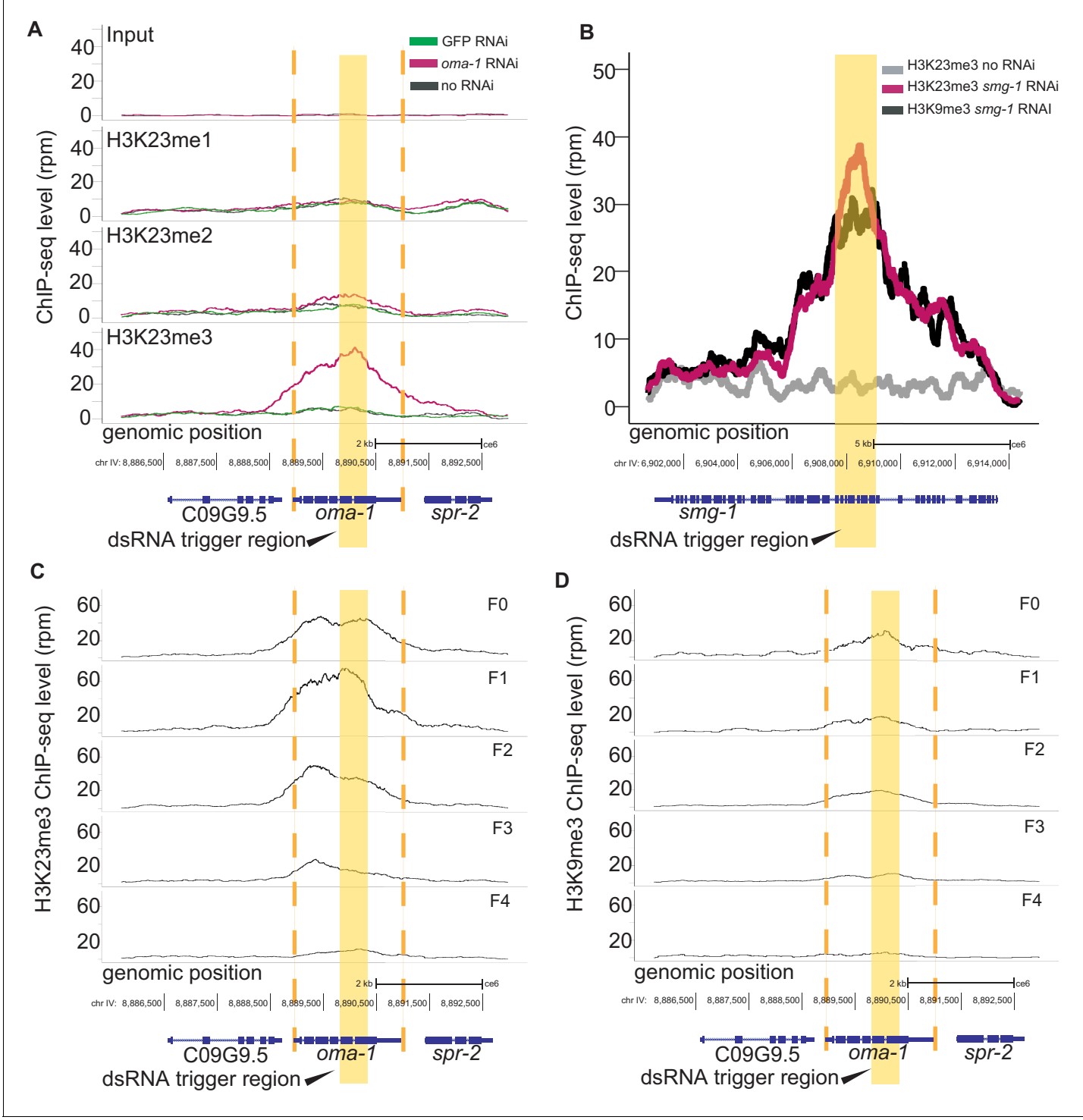

**Figure 2.** dsRNA triggers transgenerational H3K23me3 at the RNAi target gene. (**A**) H3K23 methylation levels are plotted as a function of position along the *oma-1* locus. The top panel shows input DNA for the three ChIP experiments: H3K23me1 (second panel), H3K23me2 (third panel), H3K23me3 (bottom panel); *pink*: oma-1 dsRNA, *green*: GFP dsRNA, *black*: no dsRNA feed. (**B**) H3K23me3 and H3K9me3 levels are plotted as a function of position along the *smg-1* locus after *smg-1* dsRNA feeding. Yellow block highlights dsRNA trigger region. All signals are normalized to sequencing depth. (**C-D**) *oma-1* heritable RNAi assay. H3K23me3 (*left*) compared with H3K9me3 (*right*), at *oma-1* locus with *oma-1* dsRNA feeding at the F0 generation (top panel) and no dsRNA feeding in subsequent generations, F1-F4. Yellow block highlights dsRNA trigger region, orange dashed lines indicate the boundaries of *oma-1*. The data in this figure is representative of 1 replicate (H3K23me1, H3K23me2, heritable oma-1, smg-1), and three replicates for H3K23me3 and H3K9me3 *oma-1*.

*Figure 2 continued on next page*

*Figure 2 continued*

The online version of this article includes the following figure supplement(s) for figure 2:

**Figure supplement 1.** Antibody validation.
**Figure supplement 2.** Comparison of dsRNA-mediated H3K9me3 and H3K23me3.

As nuclear RNAi is heritable and requires germline factors, we also wished to assess the whether H3K23me3 is a germline-specific histone modification. *glp-1(e2141)* mutants are defective in germ cell proliferation at the restrictive temperature (25°C). By comparing the *glp-1* mutant and WT worms, we were able to determine that nuclear RNAi-induced H3K23me3 is indeed greatly enriched in the germline (*Figure 3D*). However, H3K23me3 was still present in the *glp-1* mutant animals at both nuclear RNAi targets and globally, indicating that H3K23me3 occurs in both soma and germline.

By analyzing both exogenous (*Figure 2*) and endogenous targets (*Figure 3*), we determined that H3K23me3, like H3K9me3, is a nuclear RNAi-induced heterochromatic mark.

## *set-32* and *hrde-1* are required for nuclear RNAi-induced H3K23me3

In order to elucidate the genetic requirements of nuclear RNAi-induced H3K23me3, we performed dsRNA feeding against *oma-1* in three different mutant strains, *hrde-1, set-32* single mutant, and *met-2 set-25* double mutant, followed by H3K23me3 ChIP-seq (*Figure 4A*). As expected, *hrde-1* and *set-32* mutant worms showed greatly reduced H3K23me3 at the *oma-1* locus compared to WT. By contrast, *oma-1* RNAi in the double HMT mutant, *met-2 set-25,* and the WT animals induced similar, high levels of H3K23me3, indicating that SET-32 has a specific role in H3K23 methylation not shared by the other two HMTs (*Figure 4A*).

Interestingly, the genetic requirements at endogenous targets were harder to parse. We did not observe consistent reduction in the mutants across different endogenous targets (*Figure 4B–D* & *Figure 4—figure supplement 1D–G*). We found that while some targets display loss of H3K23me3 in *hrde-1* and *set-32* mutants but not *met-2 set-25* (*Figure 4B* and *Figure 4—figure supplement 1*), this was not true for all of them. At some loci, the reduction in H3K23me3 was the most apparent in either *set-32* mutants (*Figure 4C* and *Figure 4—figure supplement 1E*) or in *hrde-1* mutants (*Figure 4D*). Still other targets displayed a weak reduction in all three mutations (*Figure 4—figure supplement 1D*). These findings indicate that the regulation of H3K23me3 at nuclear RNAi targets is complex, and likely to involve other HMTs. When we examined the loss of H3K9me3 in the mutants at the same genomic loci, we do not see the same variation (*Figure 4—figure supplement 3*). These data indicate a complex regulation of heterochromatin marks at the endogenous targets and suggest that the genetic requirements may be contingent on local chromatin environment and other factors.

To further quantify these results, we used whole genome scatter plots (*Figure 4—figure supplement 1A–C*) and box plots (*Figure 4E*). We did not observe complete abolishment of H3K23me3 in the *set-32* or *hrde-1* mutant animals at the global level (*Figure 4—figure supplement 1A–C*). However, we did see a strong loss of H3K23me3 at endogenous nuclear RNAi targets in both the *hrde-1* and *set-32* mutant animals and weaker H3K23me3 reduction in the *met-2 set-25* double mutant animals (*Figure 4E* and *Figure 4—figure supplement 1A–C*). In published works, we have shown that H3K9me3 has a larger requirement of MET-2 and SET-25 than SET-32, further supporting the different functions of SET-32 and MET-2/SET-25 (*Kalinava et al., 2017*; *Spracklin et al., 2017*).

In order to better understand the contribution of the three HMTs and HRDE-1 to H3K23me3, we used Venn diagram analysis (*Figure 4F*). Previous studies have suggested that SET-32 is a nuclear RNAi-specific factor, we therefore expected that the regions of SET-32-dependent H3K23me3 would overlap with HRDE-1 targets. To probe this question, we identified genes with SET-32 or HRDE-1-dependent H3K23me3 profiles. The lists of the top SET-32 and HRDE-1-dependent genes are heavily populated by our previously defined endogenous targets and contain several classes of repeat elements (*Supplementary file 1* and *Figure 4—figure supplement 1F–G* for example coverage plots). We then performed a Venn diagram analysis and found that the majority of SET-32-dependent genes overlapped with HRDE-1-dependent genes (*Figure 4F*). There are over 150 SET-32-dependent H3K23me3 genes, in contrast there are only 40 MET-2 SET-25-dependent H3K23me3 genes. In

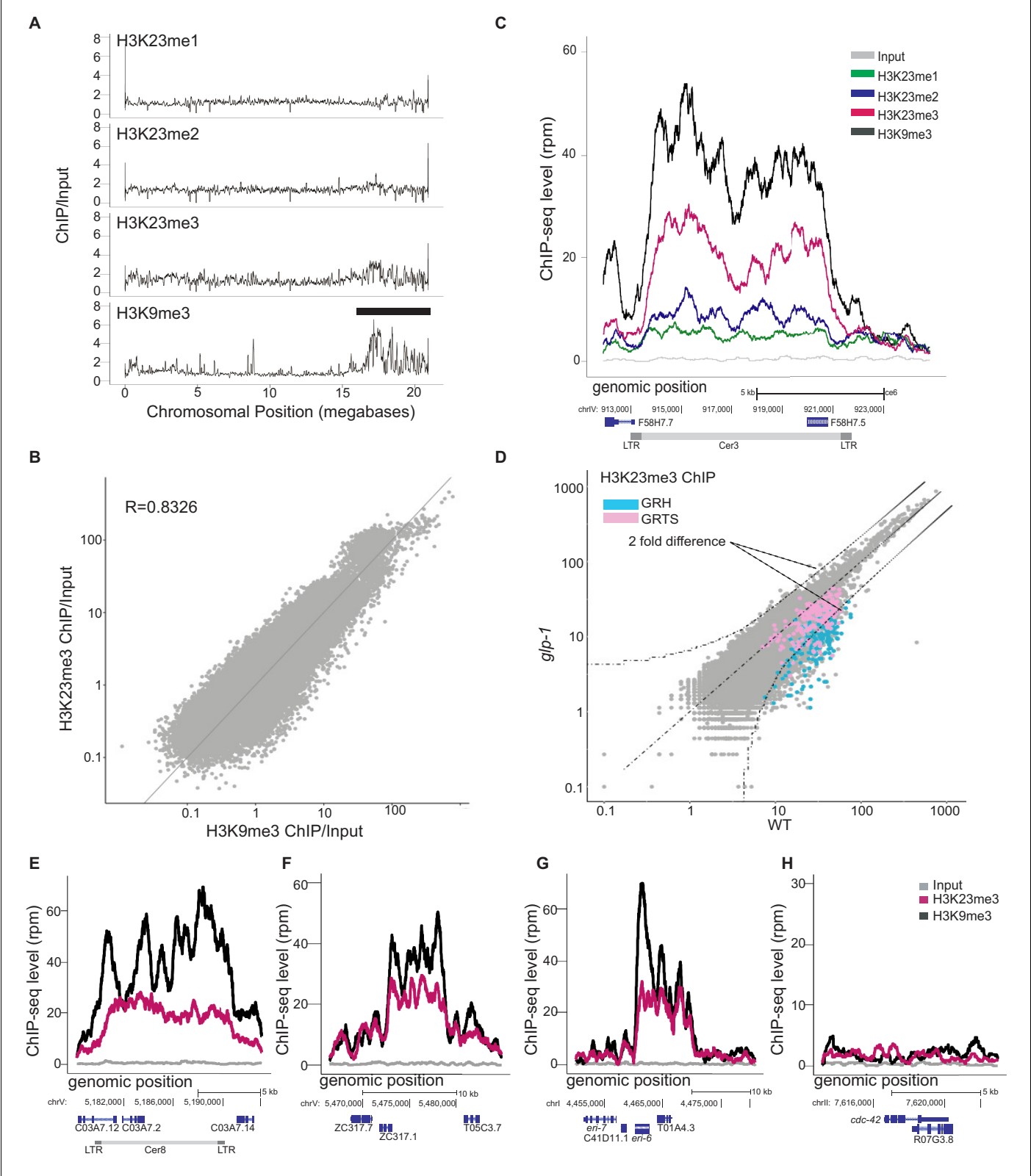

**Figure 3.** H3K23 methylation profiles at endogenous germline nuclear RNAi targets in WT. (**A**) Relative enrichment (y axis) of H3K23me1, H3K23me2, H3K23me3, H3K9me3 ChIP (top to bottom) to input for chromosome V (x axis). Black bar indicates approximate location of meiotic paring center. (**B**) Scatter plot of H3K23me3 ChIP/input (y axis) vs H3K9me3 ChIP/input (x axis) for the whole genome, in which each point represents a 1kb segment of the genome. Averaged values from two replicates were used. (**C**) H3K23me1, H3K23me2, H3K23me3, and H3K9me3 enrichment (y axis) at an

*Figure 3 continued on next page*

*Figure 3 continued*

endogenous germline nuclear RNAi target, *Cer3* LTR retrotransposon (x axis). *Grey* is the input signal. (D) Scatter plot comparing the H3K23me3 whole-genome profiles (1kb windows) in *glp-1(e2141)* and WT adult animals (25°C). Curved dashed lines indicated two-fold difference (FDR≤0.05). Regions of germline nuclear RNAi-mediated heterochromatin (GRH) are highlighted in blue and regions of germline nuclear RNAi-mediated transcriptional silencing (GRTS) in pink. (E-H) H3K9me3 (*black*) and H3K23me3 (*pink*) coverage plots for three other endogenous targets, (E) *Cer8,* (F) an exemplary GRH locus on chromosome V:5465000-5485000, (G) *eri-6,* and (H) a control euchromatin locus, *cdc-42*. All signals are normalized to sequencing depth. The data in this figure is representative of 1 replicate (H3K23me1, H3K23me2, H3K27me3, *glp-1*), or 3 replicates (H3K23me3 & H3K9me3).

The online version of this article includes the following figure supplement(s) for figure 3:

**Figure supplement 1.** Whole Chromosome Comparison of H3K9me3/H3K27me3/H3K23me3/H3K23me2/H3K23me1.

**Figure supplement 2.** Boxplot comparing relative enrichment of heterochromatic chromosome arms to chromosome center for H3K9 and H3K23 methylation.

comparison, there are three times as many MET-2 SET-25-dependent H3K9me3 genes as there are SET-32-dependent H3K9me3 genes (*Figure 4—figure supplement 2*).

To further explore the targets of H3K23me3 in nuclear RNAi, we performed gene analysis on the *hrde-1*-dependent H3K23me3 genes. In this analysis, we found that while no class of genes was enriched in these targets, 61.4% of the genes were enriched for repeat DNA of varying classes (*Figure 4—figure supplement 4A*). The majority of repeats were low complexity and DNA repeats (*Figure 4—figure supplement 4C*). However, there was also an enrichment of LTRs (4.37% percent of genes containing repeats contained LTRs) (*Figure 4—figure supplement 4C*). We did the same analysis of *hrde-1*-dependent H3K9me3 genes and found no significant difference in repeat frequency or type (*Figure 4—figure supplement 4B and D*).

## Discussion

### *C. elegans* as a model system to study H3K23 methylation

H3K23 methylation was discovered in alfalfa in 1990 (*Waterborg, 1990*) and has subsequently been found in yeast, *Tetrahymena, C. elegans*, mouse, rat, pig and human (*Waterborg, 1990*; *Garcia et al., 2007*; *Liu et al., 2010*; *Liu et al., 2013*; *Zhang et al., 2013*; *Papazyan et al., 2014*; *Vandamme et al., 2015*; *Sidoli et al., 2016b*; *Su et al., 2016*; *Fišerová et al., 2017*; *Myers et al., 2018*; *Woodhouse et al., 2018*; *Lin et al., 2020*). Despite its high degree of evolutionary conservation, H3K23me remains an understudied histone modification. H3K23 is the second most highly methylated lysine of H3 in *C. elegans* (*Papazyan et al., 2014*). The developmental dynamics of H3K23me have been characterized, either alone or in combination with other histone modifications (*Papazyan et al., 2014*; *Sidoli et al., 2016b*). The whole-genome distributions of H3K23me have been determined in this work and previous studies (*Vandamme et al., 2015*; *Sidoli et al., 2016b*). In addition, this study discovered the first biochemically validated H3K23 histone methyltransferase (HMT) and demonstrated that H3K23me3 can be experimentally induced at RNAi target genes. These advances uniquely position *C. elegans* as a powerful system to explore the regulation and function of H3K23 methylation.

### Nuclear RNAi induces multiple heterochromatin marks

Previous studies identified H3K9me3 and H3K27me3 as nuclear RNAi-mediated histone modifications (*Gu et al., 2012*; *Mao et al., 2015*). This study adds H3K23me3 to the list. Based on these results, we propose that the revised model should include the following features. (1) Different marks at nuclear RNAi targets are deposited by different HMTs: MET-2 and SET-25 for H3K9me3, MES-2 for H3K27me3, and SET-32 and unknown other HMT(s) for H3K23me3. (2) The genetic requirement of SET-32 for H3K9me3 suggests that H3K23me3 promotes H3K9me3. (3) At some of the endogenous nuclear RNAi targets, H3K9me3 and H3K23me3 appear to be mutually dependent. (4) H3K27me3 is largely independent of H3K9me3 and H3K23me3 (*Kalinava et al., 2017*). Future studies are needed to further delineate the relationship of these histone marks and determine the mechanisms by which the HMTs are recruited to target chromatin.

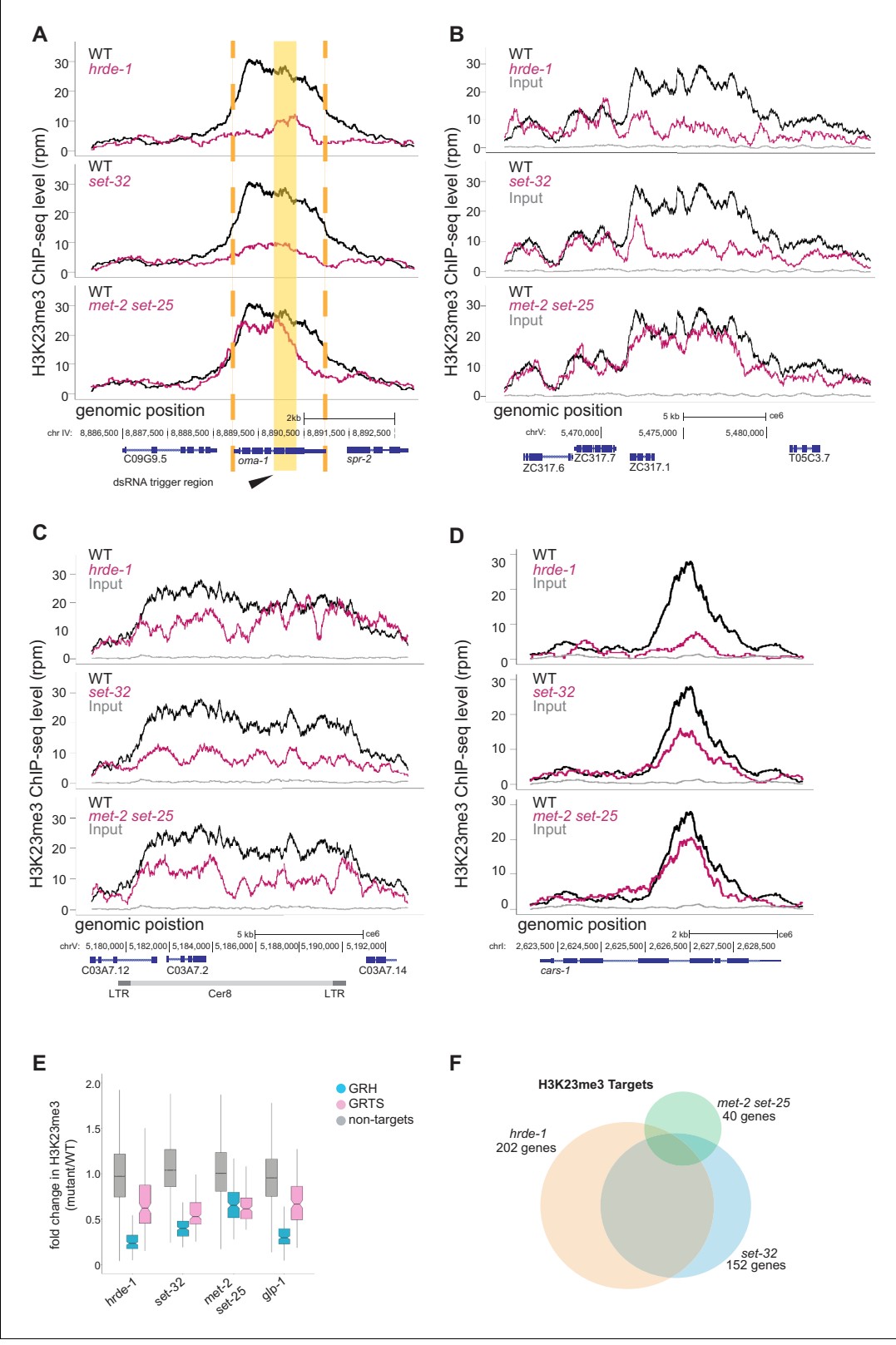

**Figure 4.** *set-32* and *hrde-1* are required for nuclear RNAi-dependent H3K23me3. (**A–D**) H3K23 methylation levels (y axis) are plotted as a function of genomic position (x axis) in three mutant strains. Top panel: *hrde-1*, middle panel: *set-32*, bottom panel: *met-2 set-25*. The same H3K23me3 signal from WT animals (*black*) was plotted in each panel to compare with the mutant signals (*pink*). Grey: ChIP input from WT. (**A**) *oma-1* locus with exogenous

*Figure 4 continued*

dsRNA-induced nuclear RNAi. Yellow block highlights the dsRNA trigger region, orange dashed lines indicate the boundaries of *oma-1*. (**B-C**) Endogenous targets of nuclear RNAi, (**B**) an exemplary GRH locus on chromosome V:5465000–5485000, (**C**) Cer8, (**D**) *Y23H5A.7a,* a top *hrde-1*-dependent gene. (**E**) Box plot of H3K23me3 ChIP whole-genome coverage in mutant/WT comparing GRH and GRTS regions, as well as the rest of the genome. (**F**) Venn diagram of genes enriched with *hrde-1* (orange), *set-32* (blue), and *met-2 set-25* (green)-dependent H3K23me3. Dependence is measured as a twofold decrease in H3K23me3 signal in mutant compared with WT for individual annotated protein-coding genes in two replicas (p value < 0.05). Fisher's exact test found the overlap between all pairs are statistically significant; the p values are < $2.2\times10^{-16}$ for *hrde-1* vs *set-32* and *set-32* vs *met-2 set-25* and $2.095 \times 10^{-15}$ for *hrde-1* vs *met-2 set-25*. The data in this figure is representative of one replicate (*hrde-1 oma-1* RNAi), two replicates (all other genotypes in *oma-1* RNAi) or three replicates (all genotypes no RNAi). The online version of this article includes the following figure supplement(s) for figure 4:

**Figure supplement 1.** Genetic requirements for H3K23me3 at additional loci.
**Figure supplement 2.** Genetic requirements for H3K9me3.
**Figure supplement 3.** Genetic requirements for H3K9me3 at additional loci.
**Figure supplement 4.** Repeat analysis of *hrde-1*-dependent H3K23me3 and H3K9me3 genes.

## What is the function of H3K23me3 in nuclear RNAi?

The involvement of SET-32 in both the maintenance and establishment phases of nuclear RNAi suggests that H3K23me3 functions in these phases as well (*Kalinava et al., 2017*; *Spracklin et al., 2017*; *Kalinava et al., 2018*; *Woodhouse et al., 2018*). During the establishment phase of nuclear RNAi, the host organism encounters a foreign genetic element for the first time and must repress its active transcription. In the maintenance phase, a stable silencing state is passed on from parent to progeny.

### Maintenance

SET-32 is dispensable for silencing maintenance when in the presence of wild-type HRDE-1. However, *set-32 hrde-1* double mutants show enhanced de-silencing when compared to *hrde-1* single mutants (*Kalinava et al., 2018*). These results suggest a conditional requirement of H3K23me3 for silencing maintenance, possibly to function as a failsafe or backup silencing mechanism.

### Establishment

Recent studies have shown that SET-32 is required for silencing establishment at both exogenous dsRNA targets and endogenous targets (*Kalinava et al., 2018*; *Woodhouse et al., 2018*). Unlike in the maintenance phase, SET-32 is required in silencing establishment, even when HRDE-1 is intact, suggesting that H3K23me3 plays a more prominent role in the establishment phase than the maintenance phase.

### The functional relationship between H3K23me3 and H3K9me3

The co-occurrence of H3K9me3 and H3K23me3 could indicate that they work together to play a larger role in heterochromatin architecture. Both H3K9me3 and H3K23me3 have been shown to bind human HP1 or its orthologs *in vitro* (*Shanle et al., 2017*). In addition, both HP1α and H3K23me2 are enriched in the nuclear pore complex in HeLa cells (H3K23me3 was not tested) (*Fišerová et al., 2017*). Future studies are needed to investigate whether similar molecular interactions occur in *C. elegans* and dissect the distinct functions associated with H3K23me3 and H3K9me3.

## Additional H3K23 histone methyltransferases

Like H3K9me3, H3K23me3 is broadly enriched in *C. elegans* heterochromatin. Our study indicates that SET-32 primarily targets the nuclear RNAi loci for H3K23me3. H3K23me3 in other heterochromatic regions is deposited by other known H3K23 HMT(s). In addition, the unknown H3K23 HMTs are likely to deposit a small fraction of H3K23me3 in the nuclear RNAi targets. We and others observed no obvious defects in animal development or mRNA expression in *set-32* mutants (*Andersen and Horvitz, 2007*; *Kalinava et al., 2017*; *Woodhouse et al., 2018*); however, given the

abundance of H3K23me in worms, we think it likely that upon identification of other H3K23me HMT (s), a combinatorial mutant will show defects. Future studies are needed to identify additional H3K23 HMTs and the broader function of H3K23me.

# Materials and methods

### Key resources table

| Reagent type (species) or resource | Designation | Source or reference | Identifiers | Additional information |
|---|---|---|---|---|
| Commercial assay or kit | KAPA Hyper Prep Kit | KAPA Biosystems | | |
| Commercial assay or kit | Vivaspin columns (MWCO 30 KDa) | GE healthcare | | |
| Peptide, recombinant protein | *Xenopus* H4 | Histone Source | | |
| Strain, strain background (*C. elegans*) | WT | Caenorhabditis Genetics Center | N2 | |
| Strain, strain background (*C. elegans*) | *set-32* | PMID:30463021 | LG I: *set-32*(red11) | |
| Strain, strain background (*C. elegans*) | *hrde-1* | Caenorhabditis Genetics Center | LG III: *hrde-1*(tm1200) | |
| Strain, strain background (*C. elegans*) | *glp-1* | Caenorhabditis Genetics Center | *glp-1*(e2141) | |
| Strain, strain background (*C. elegans*) | *met-2, set-25* | Caenorhabditis Genetics Center | *met-2*(n4256) *set-25*(ok5021) | |
| Antibody | rabbit polyclonal anti-H3K9me3 | Abcam | ab8898 | Anitbody for ChIP seq |
| Antibody | rabbit polyclonal anti-H3K23me3 | Active Motif | 61500 | Anitbody for ChIP seq |
| Antibody | rabbit polyclonal anti-H3K23me2 | Active Motif | 39654 | Anitbody for ChIP seq |
| Antibody | rabbit polyclonal anti-H3K23me1 | Active Motif | 39388 | Anitbody for ChIP seq |
| Antibody | mouse monoclonal anti-H3K27me3 | Active Motif | 39535 | Anitbody for ChIP seq |
| Antibody | Donkey Anti-Rabbit- Alexa Fluor 488 | Jackson Immuno Research Laboratories | 711-545-152 | (1:300) |
| Antibody | monoclonal Mouse-anti-tubulin | DSHB | AA4.3 | (1:250) |
| Antibody | Cy5-conjugated donkey anti-Rabbit IgG secondary antibody | Jackson Immuno Research Laboratories | 711-175-152 | (1:1000) |
| Antibody | Cy5-conjugated donkey anti-Mouse IgG secondary antibody | Jackson Immuno Research Laboratories | 715-175-150 | (1:1000) |
| Peptide, recombinant protein | histone H3K23me3 | Active Motif | 31264 | antibody validation |
| Peptide, recombinant protein | histone H3K9me3 | Active Motif | 31601 | antibody validation |
| Peptide, recombinant protein | unmodified histone H3 | Abcam | ab2903 | antibody validation |
| Peptide, recombinant protein | H3K27me3 histone peptide | Abcam | ab1782 | antibody validation |
| Plasmid | pGEX-6p-1-GST-SET-32-WT | This study | pSG361 | recombinant protein expression |
| Plasmid | pGEX-6p-1-GST-SET-32-(Y448F) | This study | pSG434 | recombinant protein expression |

*Continued on next page*

*Continued*

| Reagent type (species) or resource | Designation | Source or reference | Identifiers | Additional information |
|---|---|---|---|---|
| Plasmid | pGEX-6p-1-GST-SET-25-WT | This study | pSG355 | recombinant protein expression |
| Plasmid | pet28a_human_H3.1_WT | Addgene, a gift from Joe Landry | 42631 | recombinant protein expression |
| Plasmid | pet28a_human_H3.1_K4L | This study | pSG371 | recombinant protein expression |
| Plasmid | pet28a_human_H3.1_K9L | This study | pSG367 | recombinant protein expression |
| Plasmid | pet28a_human_H3.1_K14T | This study | pSG372 | recombinant protein expression |
| Plasmid | pet28a_human_H3.1_K18L | This study | pSG373 | recombinant protein expression |
| Plasmid | pet28a_human_H3.1_K23L | This study | pSG374 | recombinant protein expression |
| Plasmid | pet28a_human_H3.1_K27L | This study | pSG375 | recombinant protein expression |
| Plasmid | pet28a_human_H3.1_K36L | This study | pSG376 | recombinant protein expression |
| Plasmid | pet28a_human_H3.1_K37L | This study | pSG377 | recombinant protein expression |
| Plasmid | pet28a_elegans_H2A | This study | pSG395 | recombinant protein expression |
| Plasmid | pet28a_elegans_H2B | This study | pSG427 | recombinant protein expression |
| Plasmid | pet28a_elegans_H3.1 | This study | pSG428 | recombinant protein expression |
| Plasmid | L4440-oma-1 | PMID:28228846 | pSG42 | RNAi feeding plasmid against oma-1 |
| Plasmid | L4440-smg-1 | PMID:22231482 | pSG27 | RNAi feeding plasmid against smg-1 |
| Plasmid | L4440-GFP | A gift from Andrew Fire lab | L4417 | RNAi feeding plasmid against GFP |

## Plasmid construction and recombinant protein purification for GST-fusion proteins

The pGEX-6P-1-GST-SET-32-WT (pSG361) and pGEX-6P-1-GST-SET-25-WT (pSG355) were generated by inserting *set-32* and *set-25* cDNA fragments into the plasmid pGEX-6P-1 using the BamHI and NotI sites. The *set-32* cDNA fragment was amplified by RT-PCR using *C. elegans* (N2) mRNA. The codon optimized *set-25* cDNA fragment was purchased from IDT as gBlock DNA. To create the pGEX-6P-1-GST-SET-32-Y448F plasmid (pSG434), the AccI-NotI fragment of the pGEX-6P-1-GST-SET-32-WT was replaced with gBlock DNA from IDT carrying the Y448F mutation. Plasmids sequences were confirmed by Sanger sequencing. pGEX-6P-1-GST-Clr4 was a gift from the Danesh Moazed (*Iglesias et al., 2018*).

The procedure for protein expression and purification was adapted from *Iglesias et al., 2018*. Briefly, *E. coli* BL21-Gold (DE3) that was transformed with the expression plasmid was cultured in 2xYT at 37°C until OD600 reached 0.6–0.8, followed by incubating on ice for 30 min, and then 30 min of 18°C in shaker before IPTG induction. Recombinant protein expression was induced by 0.2 mM IPTG and continued overnight in the 18°C shaker. All samples and reagents were placed on ice or in the cold room during protein purification. Cells were collected, resuspended in a lysis buffer (150 mM NaCl, 20 mM sodium phosphate pH = 7.4, 1% Triton X-100, 1 mM DTT, and 1 mM PMSF), and lysed by Bioruptor using the high output and five 8-min sessions with 30 s on/30 s off cycle. A large fraction of the GST-SET-32 and GST-SET-25 were lost as inclusion bodies. After a clear spin, the soluble GST-tagged protein was pulled down by rotating the sample with glutathione sepharose

beads for 1 hr. The beads were washed in a buffer containing 150 mM NaCl, 20 mM sodium phosphate pH = 7.4, 1 mM DTT, and 1 mM PMSF three times. Protein was eluted using a buffer containing 50 mM Tris-HCl pH = 8.0, 15 mM glutathione, 10% glycerol, 1 mM PMSF, and 1 mM DTT, dialyzed against the storage buffer (50 mM Tris-HCl pH = 8.0, 10% glycerol, 1 mM PMSF, and 1 mM DTT), and concentrated using the Vivaspin columns (MWCO 30 KDa, GE healthcare). The remaining soluble GST-SET-32 aggregates to form >600 KDa complex as measured by size exclusion chromatography analysis (*Figure 1—figure supplement 1C*).

## Plasmid construction and recombinant protein purification for histone H3 proteins

Plasmid pet28a_human_H3.1 (a gift from Joe Landry, Addgene plasmid # 42631) was used to construct the mutant H3 expression plasmids used in this study. The H3K4L, H3K9L, H3K14T, and H3K18L mutations were introduced by replacing the NcoI-MscI fragment in pet28a_human_H3.1 with NcoI-MscI fragments containing the corresponding mutations. Similarly, the MscI-AgeI fragments were used to make the H3K23L and H3K27L mutations and the AgeI-SalI fragments were used for H3K36L and H3K37L mutations. The single-stranded oligoes or oligo duplex pairs were used to make the mutation-containing fragments. Plasmids pet28a_elegans_H2A (*his-12*, pSG395), pet28a_elegans_H2B (*his-11*, pSG396), pet28a_elegans_H3.1 (*his-9*, pSG397), and pet28a_elegans_H4 (*his-31*, pSG398) were constructed by inserting the cDNA fragments amplified using *C. elegans* mRNA by RT-PCR into the NcoI and NotI sites. The *C. elegans* H2A, H2B, H3.1, and H4 cDNA fragments were amplified by using mRNA isolated from wild type animals. Plasmids sequences were confirmed by Sanger sequencing.

Histone purification was performed using the protocol as previously described in *Klinker et al., 2014*. Briefly, *E. coli* BL21-Gold (DE3) that was transformed with the expression plasmid was cultured in 2xYT medium at 37°C until OD600 reached 0.6–0.8. After 1 mM IPTG was added, cells were cultured for 4 hr at 37°C. Two gams of cells were resuspended with 1 ml 10x SA buffer (400 mM sodium acetate, pH 5.2, 10 mM EDTA, 100 mM lysine), 0.5 ml 4M NaCl, 100 µl 100 mM PMSF, 100 µl HALT protease inhibitor, 3.5 µl 2-mercaptoethanol, 3.6 g urea, and diH2O to a final volume of 10 ml. Cells were lysed by Bioruptor using the high output for five 8-min sessions with 30 s on/30 s off cycle. After 20 min of spin at 41,000xg, the supernatant was filtered with 0.45 µM syringe filter and passed through the HiTrap Q 5 ml column on a FPLC machine. The flow-through was loaded onto HiTrap SP 5 ml column. The elution was done by 25 ml 0–19% buffer B and 50 ml 19–50% buffer B. Buffer A contains 40 mM sodium acetate, pH 5.2, 1 mM EDTA, 10 mM lysine, 200 mM NaCl, 6 M urea, 1 mM PMSF, 5 mM 2-mercaptoethanol. Buffer B has 1 M NaCl and is otherwise the same as buffer A. Histone containing fractions were determined by SDS-PAGE/coomassie analysis. The expression of *C. elegans* H2A, H2B, and H3 was successfully but H4 was not (*Figure 1—figure supplement 3*). *Xenopus* H4 (Histone Source), which is identical to *C. elegans* H4 except at position 74 (threonine in *Xenopus* and cysteine for *C. elegans*), was used for nucleosome assembly.

## Nucleosome assembly

Nucleosome was assembled as previously described (*Luger et al., 1999*; *Klinker et al., 2014*; *Fei et al., 2015*). Briefly, *C. elegans* H2A (12 µM), H2B (12 µM), H3 (10 µM), and *Xenopus* H4 (10 µM) (Histone Source) proteins were mixed in a 250 µl final volume containing 7 M guanidinium HCl, 20 mM Tris-HCl, pH 7.5, 10 mM DTT. Guanidinium HCl was added directly to the mixture, which was then rotated at room temperature for 30–60 min to dissolve guanidinium and spun to clear the mixture. The supernatant was dialyzed against 1L refolding buffer (2M NaCl, 10 mM Tris-HCl, pH 7.5, 1 mM EDTA, 5 mM 2-mercaptoethanol) three times (2 hr overnight, and 2 hr in the cold room). After a clear spin, the sample was fractionated by size exclusion chromatography (Superdex 200 10/300 GL) in the refolding buffer. The histone octamer fractions were identified by SDS-PAGE/coomassie analysis (*Figure 1—figure supplement 3*). 12 µg 601 DNA (Histone Source) in 2 M NaCl was mixed with 12 µg histone octamer in the refolding buffer and then dialyzed against a series of buffers with 10 mM Tris-HCl, pH 7.5, 1 mM EDTA, 1 mM DTT, and reducing amounts of NaCl (1 M, 0.8 M, 0.6 M, and 0.05 M), 2 hr for each dialysis. The reconstituted nucleosome was heated at 55°C for 20 min and then cooled to room temperature for 10 min for most thermal stable nucleosome positioning. The nucleosome was concentrated and used for the HMT assay.

## HMT assay

The HMT assay was performed as described in *Iglesias et al., 2018*. Briefly, the [3H]-based HMT assay was carried out in a 10 µl reaction mix containing 2 µM histone or nucleosome, 5.6 µM [3H]-S-adenosyl methionine (SAM), 0.3–1 µM enzyme, and 1x HMT buffer (50 mM Tris-HCl, pH 8.0, 20 mM KCl, 10 mM $MgCl_2$, 0.02% Triton X-100, 1 mM DTT, 5% glycerol, and 1 mM PMSF). The reaction mix was incubated at 20°C for 2 hr, and then loaded onto 17% SDS-PAGE. After electrophoresis, proteins were transferred to PVDF membrane, which was then soaked with autoradiography enhancer (EN3HANCE, PerkinElmer) and then air dried. Fluorography signal was detected by X-ray film. For the histone mass spectrometry analysis, 150 µl reaction mix containing approximately 0.3 µM GST-SET-32, 2.5 µM H3, and 213 µM SAM and 1 x HMT buffer without Triton X-100 was used.

## Mass spectrometry

Histone peptides were obtained as previously described (*Sidoli et al., 2016a*). Briefly, histone pellets were resuspended in 20 µL of 50 mM $NH_4HCO_3$ (pH 8.0) plus 5 µl of acetonitrile. Derivatization was performed by adding 5 µl of propionic anhydride rapidly followed by 16 µl of ammonium hydroxide and incubated for 20 min at room temperature. The reaction was performed twice to ensure complete derivatization of unmodified and monomethylated lysine residues. Samples were then dried, resuspended in 20 µL of 50 mM $NH_4HCO_3$ and digested with trypsin (Promega) (enzyme:sample ratio = 1:20, 2 hr, room temperature). The derivatization reaction was then performed again twice to derivatize peptide N-termini. Samples were then desalted by using in-house packed stage-tips and dried using a SpeedVac centrifuge.

Dried samples were resuspended in 0.1% trifluoroacetic acid (TFA) and injected onto a 75 µm ID x 25 cm Reprosil-Pur C18-AQ (3 µm; Dr. Maisch GmbH, Germany) nano-column packed in-house using a Dionex RSLC nanoHPLC (Thermo Scientific, San Jose, CA, USA). The nanoLC pumped a flow-rate of 300 nL/min with a programmed gradient from 5% to 28% solvent B (A = 0.1% formic acid; B = 80% acetonitrile, 0.1% formic acid) over 45 min, followed by a gradient from 28% to 80% solvent B in 5 min and 10 min isocratic at 80% B. The instrument was coupled online with a Q-Exactive HF (Thermo Scientific, Bremen, Germany) mass spectrometer acquiring data in a data-independent acquisition (DIA) mode as previously optimized (*Sidoli et al., 2015*). Briefly, DIA consisted on a full scan MS (*m/z* 300–1100) followed by 16 MS/MS with windows of 50 *m/z* using HCD fragmentation and detected all in the orbitrap analyzer.

DIA data were searched using EpiProfile 2.0 and validated manually (*Yuan et al., 2018*). The histone H3 peptide KQLATKAAR (aa 18–26) was considered in all possible modified forms (unmodified, me1/2/3). The relative abundance of each form was calculated using the total area under the extracted ion chromatograms of all peptides in all the (un)modified forms and considered that as 100%. To confirm the position of the methylation, we extracted the chromatographic profile of the MS/MS fragment ions and verified that no unique fragment ions belonging to the K18me1/2/3 possible peptide isoforms had detectable intensity.

## *C. elegans* strains

*C. elegans* strain N2 was used as the standard WT strain. Alleles used in this study were: LG I: *set-32* (red11), LG III: *hrde-1*(tm1200), *glp-1*(e2141), *set-25*(n5021), *set-32*(ok1457), *met-2*(n4256) *set-25* (ok5021). *C. elegans* were cultured on NMG agar plates as previously described (*Brenner, 1974*) in a temperature-controlled incubator.

## Preparation of worm grinds

Synchronized young adult worms were first washed off the plates with M9 buffer. *E. coli* OP50 bacteria washed off together with the worms were separated and removed by loading the worms to 10% sucrose cushion and centrifuging for 1 min at 600 g in a clinical centrifuge. Worms were then pulverized by grinding in liquid nitrogen with a pre-chilled mortar and pestle and were stored at −80°C.

## dsRNA feeding

Worms were grown on NGM plates with the following food sources: OP50 *E. coli* (no RNAi), *oma-1 E. coli* (plasmid SG42), GFP *E. coli* (plasmid SG221), *smg-1 E. coli* (plasmid SG27). Worms were synchronized by bleaching, subsequent starvation and released at the L1 stage onto described plates.

Each assay in this study used grinds of ~5000 worms for each condition. For RNAi experiments, worms were grown for three to four generations on RNAi culture before grinding. For heritable RNAi experiments, we used the protocol described in *Gu et al., 2012*. Briefly, worms were raised on oma-1 RNAi plates for three generations before synchronization and release onto OP50 *E. coli* plates without dsRNA feed. Worms were collected at P0, F1, F2, F3, F4 generations for grinding.

## ChIP-seq library construction

Worm grinds from approximately 5000 worms were used for each chromatin immunoprecipitation experiment according to the procedure described in *Ni et al., 2014*. Anti-H3K9me3 (ab8898, Abcam) and anti-H3K23me3 (61500, Active Motif) antibodies were used for the H3K9me3 and H3K23me3 ChIP, respectively. Each ChIP experiments usually yielded 5–10 ng DNA. The entire ChIP DNA or 10 ng DNA in the case of ChIP input was used to make DNA library with the KAPA Hyper Prep Kit (KAPA Biosystems) according to the manufacturer's instruction. For each sample in a given assay, worm grinds were thawed and subsequently crosslinked and sonicated to produce fragments between 200 and 500 bp according to protocol described in *Ni et al., 2016*. Samples were then used for ChIP (H3K23me1,2,3, H3K9me3, H3K27me3) or stored for library prep as input DNA. IP was performed with the following antibodies: anti-H3K9me3 (ab8898, Abcam), anti-H3K23me1 (39388, Active Motif), anti-H3K23me2 (39654, Active Motif), anti-H3K23me3 (61500, Active Motif), and anti-H3K27me3 (39535, Active Motif). For each antibody, ~0.5–1.5% of input DNA was pulled down, with DNA yields between ~5 and 25 ng. 5 ng or less of DNA was used for library preparation using KAPA Hyper Prep Kit (KAPA Biosystems) according to the manufacturer's instruction. PCR was performed on library DNA for 12–17 cycles after which all libraries were pooled according to Illumina HiSeq specifications. Sequencing was sent to Illumina and carried out according to the following specifications: 50-nt single-end run, dedicated index sequencing. Dedicated 6-mer indexes were used to demultiplex the libraries of different samples. All libraries used in this study are listed and described in *Supplementary file 2*.

## Antibody validation

### Immunofluorescence staining

Adult worms were washed in 1X PBS twice and paralyzed in 0.1 mM levamisole in 1X PBS. Paralyzed worms were transferred to a cavity slide and gonads were dissected using two 25-gauge syringe needles. The dissected gonads were first fixed in 100% methanol in 1X PBS at −20°C for 1 min, and then were fixed in 2% paraformaldehyde in 1X PBS at room temperature for 5 min. After fixation, gonads were blocked in blocking buffer (1 mg/ml BSA, 10% Normal Goat Serum, 0.1% Tween 20, 1X PBS) for 45 min at room temperature. For primary antibody staining, gonads were incubated in Rabbit-anti-H3K23me3 (1:150, Active Motif, 61499) in blocking buffer at 4°C overnight. For antibody competition, primary antibody was pre-absorbed with 25 ng/μL corresponding histone proteins or histone peptide at room temperature for 1 hr, and was centrifuged at 14,000 rpm at 4°C for 10 min to remove immune complexes. The histone proteins and peptide used for antibody competition were histone H3K23me3 (Active motif, 31264), histone H3K9me3 (Active motif, 31601), unmodified histone H3 (Abcam ab2903), H3K27me3 histone peptide (Abcam ab1782). For secondary antibody staining, gonads were washed three times for 5 min each wash in 1X PBS/0.1% Tween 20, and then were incubated in Donkey Anti-Rabbit-Alexa Fluor 488 (1:300, Jackson ImmunoResearch Laboratories, 711-545-152) in blocking buffer at room temperature for 2 hr. The gonads were then washed three times for 5 min each wash in 1X PBS with 0.1% Tween 20. 100 ng/ml DAPI was added to the last wash to stain chromosomes. Gonads were mounted in Slowfade (Invitrogen) onto a freshly made 2% agarose pad for imaging. Gonads were imaged using a Zeiss Axio Imager M2 system. Images were processed with Fiji (ImageJ) (*Schindelin et al., 2012*).

### Western blot

Worm grinds were lysed in 2X Laemmli buffer with 1X HALT protease and phosphatase inhibitor (Thermo Fisher Scientific) by boiling at 95°C for 5 min. Worm lysate (25 μg /lane) were loaded and separated on Bio-Rad TGX Any KD gel and transferred to nitrocellulose membrane. Primary antibodies used are polyclonal Rabbit-anti-H3K23me1 (1:1000, Active Motif 39388), polyclonal Rabbit-anti-H3K23me2 (1:1000, Active Motif 39654), polyclonal Rabbit-anti-H3K23me3 (1:1000, Active Motif

61500), polyclonal Rabbit-anti-H3K9me3 (1:1000, Abcam ab8898), monoclonal Mouse-anti-tubulin (1:250, DSHB AA4.3). Secondary antibodies used are Cy5-conjugated donkey anti-Rabbit IgG secondary antibody (1:1000, Jackson ImmunoResearch Laboratories 711-175-152) and Cy5-conjugated donkey anti-Mouse IgG secondary antibody (1:1000, Jackson ImmunoResearch Laboratories, 715-175-150). Fluorometric detection and measurement was preformed using GE/Amersham Typhoon RGB scanner and ImageQuant software (GE Healthcare).

## High-throughput sequencing

Pooled libraries were sequenced on an Illumina HiSeq 2500 platform (rapid run mode, 50-nt single-end run, and index sequencing). De-multiplexed raw data in fastq format were provided by the sequencing service facility. Library information is listed in *Supplementary file 2*. High-throughput sequencing data generated for this study have been deposited in NCBI Gene Expression Omnibus (accession number GSE141347).

## Data analysis

Fastq files of each library were aligned directly to *C. elegans* genome (WS190 version) using the Bowtie alignment program (version 1.2.2), only perfect alignments were reported and used (*Langmead et al., 2009*). When a read aligned to multiple loci the alignment was counted as 1/ (number alignments). For all data analysis, normalization was based on the sequencing depth of each library (total number of reads aligned). For some figures, normalization was additionally based on the respective input library (no antibody), where this is done it is stated in the figure legend. For individual loci coverage, each read was extended by 500 bp from the sequenced end. Whole chromosome coverage analysis was done based on 1 kb windows for the entire genome. All coverage plot figures were created using custom python scripts and custom R scripts in ggplot2.

We and others have characterized over 150 genomic loci in *C. elegans* that are de-silenced and/ or lose repressive chromatin modifications in nuclear RNAi-deficient mutants, the so-called 'endogenous targets'. These endogenous targets are also referred to as GRH and GRTS loci. GRH regions are defined as regions with germline nuclear RNAi dependence. GRTS regions are defined at regions with germline nuclear-RNAi-dependent transcriptional silencing. (*Ni et al., 2014*; *Ni et al., 2016*; *Kalinava et al., 2017*).

## ChIP-seq replicates

Plots of ChIP-seq data are representative of either one, two or three replicates as follows:

| ChIP | Genotype | RNAi | Number of replicates |
|------|----------|------|----------------------|
| H3K23me3 | N2 | *oma-1* | 2 |
| H3K23me3 | *hrde-1* | *oma-1* | 1 |
| H3K23me3 | *set-32* | *oma-1* | 2 |
| H3K23me3 | *met-2 set-25* | *oma-1* | 2 |
| H3K23me3 | N2 | none | 3 |
| H3K23me3 | *hrde-1* | none | 3 |
| H3K23me3 | *set-32* | none | 3 |
| H3K23me3 | *met-2 set-25* | none | 3 |
| H3K23me3 | *glp-1* | none | 1 |

Library information can be found in *Supplementary file 2*.

## Venn diagram analysis

We used venn.js (https://github.com/benfred/venn.js/), a library in D3.js. To layout each Venn diagram proportional to the input sizes, we defined the sets, and specified the size of each individual set as well as the size of all set intersections. The sizes of all sets and intersections were found in custom R scripts. Each set was defined using two replicate libraries.

## Repeat analysis

Repeat masker data were downloaded from UCSC table browser, with repeat class specified for each repeat. The *hrde-1*-dependent H3K23me3/H3K9me3 genes were found by comparing N2 with *hrde-1* mutant (two fold cutoff and p-value<0.05). Two replicas of *hrde-1* mutant were used and only genes that were identified in both replicas were then used for subsequent repeat analysis. Data were then saved in bed format, with chromosome start and end position for each gene or repeat.

The Bioconductor package 'ChIPpeakAnno' was used for overlapping repeats with genes (Chapter 4.1 in ChIPpeakAnno user's guide: https://www.bioconductor.org/packages/devel/bioc/vignettes/ChIPpeakAnno/inst/doc/ChIPpeakAnno.html). We used findOverlapsOfPeak (bed1, bed2) command to overlap the target genes with repeat masker tracks. The output 'peaklist' object included information regarding genes that did not overlap with any repeats. Pie charts were made to represent the number of genes that did or did not contain repeats identified by Repeat Masker. For the genes that did contain repeats we used pie charts to display what kinds of repeat classes were present in the set.

## Acknowledgements

We thank Danesh Moazed, Ruth Steward, James Millonig, Michael Verzi, and Vincenzo Pirrotta for help and suggestions. Research reported in this publication was supported by the Rutgers Busch Biomedical Grant to SSG, the National Institute of General Medical Science of the NIH, United States, under award R01GM111752 to SSG and the New Jersey Commission on Cancer Research under award DCHS19PPC030 to LSO. The content is solely the responsibility of the authors and does not necessarily represent the official views of the NIH.

## Additional information

### Funding

| Funder | Grant reference number | Author |
| --- | --- | --- |
| National Institutes of Health | R01GM111752 | Sam G Gu |
| Rutgers BuschBiomedical | | Sam G Gu |
| New Jersey Commission on Cancer Research | DCHS19PPC030 | Lianna Schwartz-Orbach |

The funders had no role in study design, data collection and interpretation, or the decision to submit the work for publication.The content is solely the responsibility of the 291 authors and does not necessarily represent the official views of the NIH.

### Author contributions

Lianna Schwartz-Orbach, Conceptualization, Resources, Data curation, Software, Formal analysis, Funding acquisition, Validation, Investigation, Visualization, Methodology, Writing - original draft, Writing - review and editing; Chenzhen Zhang, Data curation, Software, Formal analysis, Investigation, Visualization, Methodology, Writing - review and editing; Simone Sidoli, Resources, Data curation, Software, Formal analysis, Validation, Investigation, Visualization, Methodology, Writing - review and editing; Richa Amin, Diljeet Kaur, Anna Zhebrun, Investigation, Methodology; Julie Ni, Conceptualization, Validation, Investigation, Visualization, Methodology, Writing - review and editing; Sam G Gu, Conceptualization, Resources, Data curation, Software, Formal analysis, Supervision, Funding acquisition, Validation, Investigation, Visualization, Methodology, Writing - original draft, Project administration, Writing - review and editing

### Author ORCIDs

Lianna Schwartz-Orbach (iD) https://orcid.org/0000-0001-5938-7762
Sam G Gu (iD) https://orcid.org/0000-0002-5861-3630

**Decision letter and Author response**
Decision letter https://doi.org/10.7554/eLife.54309.sa1
Author response https://doi.org/10.7554/eLife.54309.sa2

## Additional files

### Supplementary files

• Supplementary file 1. Top *set-32* and *hrde-1* dependent genes. A list of top *set-32*-dependent H3K23me3 genes taken from annotated list of 20911 *C. elegans* genes. In the second list, the top *hrde-1*-dependent H3K23me3 genes. The top genes were defined as those that were threefold higher in WT versus mutant in two libraries with a p-value of <0.05.

• Supplementary file 2. List of experiments, libraries, sequencing depth used in this study. ChIP-seq libraries are listed by experiment.

• Supplementary file 3. Oligonucleotides and other sequences used in this study.

• Transparent reporting form

### Data availability

Sequencing data have been deposited in GEO under accession codes GSE141347 Mass spec raw files generated with mass spectrometry are freely accessible at https://chorusproject.org/, project number 1636 (access to the data requires creating a free account).

The following datasets were generated:

| Author(s) | Year | Dataset title | Dataset URL | Database and Identifier |
|---|---|---|---|---|
| Schwartz-Orbach L, Ni J, Gu S | 2019 | C. elegans nuclear RNAi factor SET-32 is an H3K23 methyltransferase and deposits the transgenerational heritable modification of H3K23me3 | https://www.ncbi.nlm.nih.gov/geo/query/acc.cgi?acc=GSE141347 | NCBI Gene Expression Omnibus, GSE141347 |
| Sidoli S, Gu S | 2019 | C. elegans nuclear RNAi factor SET-32 is an H3K23 methyltransferase and deposits the transgenerational heritable modification of H3K23me3 | https://chorusproject.org/pages/dashboard.html#/projects/all?q=1636 | Chorus, 1636 |

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
