## [Decision Letter]

Thank you for submitting your article "*C. elegans* nuclear RNAi factor SET-32 deposits the transgenerational heritable histone modification, H3K23me3" for consideration by *eLife*. Your article has been reviewed by Kevin Struhl as the Senior Editor, a Reviewing Editor, and three reviewers.

The reviewers have discussed the reviews with one another and the Reviewing Editor has drafted this decision to help you prepare a revised submission.

The three reviewers are interested in your manuscript and actively discussed that more functional analysis in needed to further substantiate the study regarding the role of H3K23me3 in regulation of gene expression. Furthermore, it was discussed that case inheritance of H3k23me3 is not convincing. It remains possible that small RNAs are inherited and the H3K23me3 in future generations could be a consequence. We therefore encourage you to submit a revised version of the manuscript providing more mechanistic insights in the role of small RNAs in the inheritance of H3K23me3.

Essential revisions:

In particular, we would like you to provide experimental evidence whether the H3K23me3 is inherited meiotically or re-established at each generation with the help of small RNAs. This could be done, for example by employing HRDE-1 mutants. Also, the physiological role of the H3K23me3 remains unclear. This could be address for example by performing qPCR experiments at the *oma-1* locus after 4 generations. More generally, the phenotype of the embryo lacking SET-32 should be analyzed and discussed.

The individual reviews are attached below to so that you can see the specific concerns raised by each reviewer.

Reviewer #1:

I enjoyed this very interesting paper by Schwartz-Orbach et al. It continues their past work on the involvement of HMTs and other chromatin marks (mostly K9) in silencing and transgenerational RNAi in *C. elegans*. K9 methylations, while originally thought to be crucial for heritable RNAi, are now known to be important only in specific cases (for heritable regulation of specific genes, mostly newly evolved genes, and K9 mutants are largely ok. Chromatin marks, like K9me1/2/3 do play a role in transgenerational RNAi, modulating it and controlling its duration, and this likely involves also interactions with other histone modifications. The discovery presented here, that H3K23me3 (an abundant mark) plays a role in nuclear RNAi, could shed light on the mechanism. The authors provide evidence that their antibody is specific, and that loci which are targeted by endogenous and exogenous small RNAs indeed get modified with the mark. Also, the involvement of SET-32 and HRDE-1 (both play a role but are not completely required) is convincing. I think these are all valuable discoveries that would be interesting to many in the field.

Reviewer #2:

In the paper "*C. elegans* nuclear RNAi factor SET-32 deposits the transgenerational heritable histone modification, H3K23me3" the authors use *C. elegans* as a model system to study the H3K23me3 epigenetic mark in the germline.

The current work in *eLife* from Sam Gu lab is really timing and potentially could increase our understanding of H3K23me3, its role in the transgenerational inheritance of repressive epigenetic states and its role in the phenomenon of RNAi. Although the paper is very sound, there are some major points below that need to be addressed and that are required to strengthen the paper for a publication in *eLife*.

From their study, it is clear that SET-32 methylation activity is toward H3K23 and that the appearance of H3K23me3 relies upon exogenous dsRNA. Despite the authors show in Figure 2B that the levels of H3K23me3 seems higher than H3K9me3 (after 4 generations), I believe this is not very convincing and I would rather interpret from their plot that H3K23me3 behave very similarly to H3K9me3 (upon normalisation). This data should be validated by qChIP, just on this locus.

The authors should also tag SET-32 so that to perform ChIP-seq and also IP from whole embryo with subsequent mass spec, trying to identify the protein complex containing SET-32.

The paper also lacks some more functional studies. The authors should deplete/ or make a catalytic dead of SET-32 and study the phenotype of the worms. Moreover, in this mutant background the authors should study the transcriptional effect/defect on exogenous and endogenous RNAi depended sites.

Reviewer #3:

This is quite an interesting study that characterises SET-32 as an H3K23me3 histone methyl transferase and links this under-studied histone modification to the nuclear RNAi pathway and heterochromatin in *C. elegans*.

What is missing is any convincing insight into the role of H3K23me3 in the nuclear RNAi pathway or gene repression. Therefore, the speculations in the discussion trying to link published SET-32 phenotypes to the role of H3K23me3 in the establishment or maintenance of nuclear RNAi mediated silencing are very vague. Otherwise the overall quality of the study seems to be good.

Sometimes the statements in the text do not reflect the data (e.g. ChIP data profiles subsection “*set-32* and *hrde-1* are required for nuclear RNAi-induced H3K23me3”. vs Figure 4—figure supplement 1).

Essential revisions:

It is hard to untangle the SET-32 H3K23 and H3K9 HMT activity – one possibility would be to create mutants that lack one or the other HMT activity. Also, site-specific (e.g. *oma-1* locus) demethylation of H3K9me3 by Cas9-based epigenome editing. Both approaches are however probably out of scope for this study. *set-32* mutants still show H3K23me3 marks, which speaks for the presence of additional and unknown H3K23me3 HMT(s), making functional insights even more challenging to obtain. RNA expression data to accompany the multigenerational ChIP profiles at the *oma-1* locus (Figure 2B). This would reveal whether there is still a significant repression at the F4 generation when H3K9me seems to be reduced to background levels, but H3K23me3 still shows a (small) peak.

Figure 4B does not appear to be a very representative locus to show given the genome-wide data but rather one that matches the prior hypothesis.

The exact definitions of GRH and GRTS genes need to be given in the methods.

Subsection “H3K23me3 is a heterochromatic mark in *C. elegans”*. referring to Figure 3A: there is similarity between H3K9me3 and H3K23me3; however, such similarity is not present between H3K9me3 and H3K23me2 as stated by the author. Similarity on different chromosomes (Figure 3—figure supplement 1) also shows a very weak correlation between H3K9me3 and H3K23me3 at Chr III.

Subsection “H3K23me3 is a heterochromatic mark in *C. elegans”* states that H3K23me1 is uniformly distributed (Figure 3—figure supplement 1) but it looks like H3K23me2 distribution across the genome is just as uniform and doesn't cluster at the constitutive heterochromatic regions as H3K23me3 or H3K9me3.

These results and distributions all need quantifying.

Results section: There is no reference to Figure 4B in the text (endogenous *hrde-1* target).

Results section: Figure 4 should be renamed Figure 4C-G as these represent the genome-wide data mentioned in the sentence.

Results section: Figure 4—figure supplement 1 shows slightly stronger H3K9me3 reduction in regions of germline nuclear RNAi-mediated heterochromatin (GRH). However, regions of germline nuclear RNAi-mediated transcriptional silencing (GRTS) show even a stronger depletion of H3K9me3 in *set-32* mutants compared to *met-2 set-25* double mutants. This is not in line with the statement of the author "As in published works, we found that H3K9me3 has a larger requirement of MET-2 and SET-25 than SET-32" and the conclusion that this further supports different functions of SET-32 and MET-2/SET-25.

[Editors' note: further revisions were suggested prior to acceptance, as described below.]

Thank you for submitting your article "*C. elegans* nuclear RNAi factor SET-32 deposits the transgenerational heritable histone modification, H3K23me3" for consideration by *eLife*. Your article has been reviewed by Kevin Struhl as the Senior Editor, a Reviewing Editor, and three reviewers. The reviewers have opted to remain anonymous.

The reviewers have discussed the reviews with one another and the Reviewing Editor has drafted this decision to help you prepare a revised submission.

Reviewers are overall supportive and also sympathetic about the circumstances based on which some of the additional key experiments could not be performed. However, since you have not addressed the most significant concerns of the reviewers, namely that the multi-generation 'inheritance' of H3K23me3 may be a secondary consequence of the transmission of small RNAs or other histone modifications and the lack of 'function' for H3K23me3. The title and associated text need to be toned down.

Furthermore, please explicitly state (Figure legend and text) where experiments are performed without biological replicates and tone down the strength of the conclusions. For example, as pointed out by reviewer#2: "These findings indicate that the regulation of H3K23me3 at nuclear RNAi targets is complex, and likely to involve other HMTs", as this cannot be concluded based on experiments lacking biological replicates and therefore not being quantitative.

Reviewer #1:

I didn't have too many issues with the original submission and the few questions that I had have been answered in the revision process. The authors wrote that their attempts to examine the role of HRDE-1 were disrupted due to COVID-19, that's reasonable.

Reviewer #2:

The authors have addressed some major concerns and the paper could be accepted for publication.

I, though, noticed that many ChIP experiments do not have biological replicates.

This is especially relevant for the second paragraph of subsection “*set-32* and *hrde-1* are required for nuclear RNAi-induced H3K23me3”; where the authors try to understand the loss of H3K23me3 in different mutant backgrounds.

I suggest to remove from this paragraph the sentence " These findings indicate that the regulation of H3K23me3 at nuclear RNAi targets is complex, and likely to involve other HMTs.", as this cannot be concluded based on experiments lacking biological replicates and therefore not being quantitative.

The authors should also mention in the figure legends the number of biological replicates for each experiment, each time, as otherwise for the reader it is hard to find this info.

Reviewer #3:

Unfortunately the authors have not been able to definitively address the most important referee concerns – that the multi-generation 'inheritance' of H3K23me3 may be a secondary consequence of the transmission of small RNAs or other histone modifications and the lack of 'function' for H3K23me3. However, I do think this is an interesting body of work.

For the reasons stated in the original reviews, I think the title has the potential to be misleading to a general audience: 'transgenerational heritable histone modification' is a strong statement and something that has not been directly demonstrated and I think this should be toned down.

---

## [Author Response]

Essential revisions:In particular, we would like you to provide experimental evidence whether the H3K23me3 is inherited meiotically or re-established at each generation with the help of small RNAs. This could be done, for example by employing HRDE-1 mutants.

We agree that this is an important question in the field of nuclear RNAi. We have been developing auxin-inducible degradation (AID) of HRDE-1 to address this question. To prepare for the revision, we constructed and performed some initial characterization of the AID system, which had produced promising preliminary results. Unfortunately, the experiments were interrupted by the pandemic lockdown. Given the uncertainty of the situation and the scope of the project, we feel that the HRDE-1 AID experiments are more suitable for a future paper.

Also, the physiological role of the H3K23me3 remains unclear.

Our study indicated that SET-32 is not the only writer for H3K23me. Understanding the physiological role of H3K23me3 requires the identification of the other writers, as well potential H3K23me readers, which is beyond the scope of this study. As suggested by reviewer #2, characterizing a catalytic inactive SET-32 mutant may provide connection between H3K23me3 and the phenotype of the *set-32*(-) (e.g. silencing establishment defect). This turned out to be more challenging than expected as all of our five attempts of creating the mutant failed.

This could be address for example by performing qPCR experiments at the oma locus after 4 generations.

It is well established that heritable RNAi of *oma-1* can persist for at most 4-5 generations in WT animals (e.g. PMIDs:18757930, 28343968, 32348780). We and others have shown that SET-32 promotes silencing establishment, presumably through H3K23me3, but is dispensable for heritable silencing at *oma-1*(PMIDs:30463020 and 30463021). The function of H3K23me3 (and also H3K9me3) in the heritable generations is unclear at this point.

More generally, the phenotype of the embryo lacking SET-32 should be analyzed and discussed.

We performed brood size analysis and found that *set-32*(-) animals have similar brood sizes as WT animals at both 20ºC (WT:303, *set-32*(-):307) and 25ºC (WT:203, *set-32*(-):219). In addition, a published work (PMID 30463020) did not observe any developmental defects in *set-32*(-) animals, including embryogenesis, we have included these points in the revised manuscript (Discussion section).

Reviewer #2:[…] Although the paper is very sound, there are some major points below that need to be addressed and that are required to strengthen the paper for a publication in eLife.From their study, it is clear that SET-32 methylation activity is toward H3K23 and that the appearance of H3K23me3 relies upon exogenous dsRNA. Despite the authors show in Figure 2B that the levels of H3K23me3 seems higher than H3K9me3 (after 4 generations), I believe this is not very convincing and I would rather interpret from their plot that H3K23me3 behave very similarly to H3K9me3 (upon normalisation). This data should be validated by qChIP, just on this locus.

We agree that comparison of the relative enrichment of different histone marks, needs to be careful. Both H3K23me3 and H3K9me3 at the F4 are much lower than the P0 levels. In the revised manuscript, we removed the description regarding the comparison of H3K9me3 and H3K23me3 levels at the F4 generation and put H3K9me3 and H3K23me3 in different panels (2C and 2D) to avoid confusion.

The authors should also tag SET-32 so that to perform ChIP-seq and also IP from whole embryo with subsequent mass spec, trying to identify the protein complex containing SET-32.

We have performed size exclusion chromatography analysis of SET-32 from the crude worm lysate. All the detectable SET-32 behaved as a monomer in this analysis (data not shown). The result suggests that either (1) SET-32 functions as a monomer, (2) the interaction between SET-32 and its binding partner is weak, or (3) only a very small fraction of SET-32 function as a complex. Distinguishing these possibilities are beyond the scope of this paper.

The paper also lacks some more functional studies. The authors should deplete/ or make a catalytic dead of SET-32 and study the phenotype of the worms. Moreover, in this mutant background the authors should study the transcriptional effect/defect on exogenous and endogenous RNAi depended sites.

As mentioned above, we agree with the importance of studying catalytic dead *set-32* mutant. However, we were not able to obtain the mutation after trying five different CRISPR designs.

Reviewer #3:[…]What is missing is any convincing insight into the role of H3K23me3 in the nuclear RNAi pathway or gene repression. Therefore, the speculations in the discussion trying to link published SET-32 phenotypes to the role of H3K23me3 in the establishment or maintenance of nuclear RNAi mediated silencing are very vague.

We agree that functional insight is lacking, as mentioned above, our study indicated that SET-32 is not the only writer for H3K23me. Understanding the role of H3K23me3 requires the identification of the other writers, as well potential H3K23me readers, which is beyond the scope of this study.

Otherwise the overall quality of the study seems to be good.Sometimes the statements in the text do not reflect the data (e.g. ChIP data profiles subsection “set-32 and hrde-1 are required for nuclear RNAi-induced H3K23me3” vs Figure 4—figure supplement 1).

We have updated the text in several places to better reflect the data. Thank you for alerting us to these oversights.

Essential revisions:It is hard to untangle the SET-32 H3K23 and H3K9 HMT activity – one possibility would be to create mutants that lack one or the other HMT activity. Also, site-specific (e.g. oma-1 locus) demethylation of H3K9me3 by Cas9-based epigenome editing. Both approaches are however probably out of scope for this study.

We agree that both questions are highly interestingly. In our *in vitro* biochemical analysis, the recombinant SET-32 only has H3K23 HMT activity and does not methylate any other lysines in H3. The amino acid sequence flanking K9 and K23 are distinct, therefore, it’s unlikely that SET-32 can also methylate H3K9 *in vivo* although this has not been formally ruled out. Future study is required to untangle the functional relationship between H3K9me3 and H3K23me3. We have included these points in the Discussion section of the revised manuscript.

Site-specific demethylation of H3K9me3 or H3K23me3 by Cas9-based epigenome editing is a very interesting idea. We are not aware of this approach having been successfully used in *C. elegans*. We agree that building such a tool is indeed out of scope for this study.

set-32 mutants still show H3K23me3 marks, which speaks for the presence of additional and unknown H3K23me3 HMT(s), making functional insights even more challenging to obtain. RNA expression data to accompany the multigenerational ChIP profiles at the oma-1 locus (Figure 2B). This would reveal whether there is still a significant repression at the F4 generation when H3K9me seems to be reduced to background levels, but H3K23me3 still shows a (small) peak.

We responded to this issue above.

In addition, as suggested by reviewer #2, it is difficult to compare relative enrichment of two different histone marks. Both H3K9me3 and H3K23me3 at the F4 were much weaker than the P0 levels. We have removed the description regarding the comparison between H3K9me3 and H3K23me3 at F4.

Figure 4B does not appear to be a very representative locus to show given the genome-wide data but rather one that matches the prior hypothesis.

We have added three more loci to Figure 4 to better represent the data. In addition, there are four other loci plotted in Figure 4—figure supplement 1. We have updated the text to further explain the differences in reduction of the same mutants at different loci. We expanded the descriptions of these findings to make it clearer.

The exact definitions of GRH and GRTS genes need to be given in the methods.

We have added this in the revised manuscript.

Subsection “H3K23me3 is a heterochromatic mark in C. elegans”. referring to Figure 3A: there is similarity between H3K9me3 and H3K23me3; however, such similarity is not present between H3K9me3 and H3K23me2 as stated by the author. Similarity on different chromosomes (Figure 3—figure supplement 1) also shows a very weak correlation between H3K9me3 and H3K23me3 at Chr III.Subsection “H3K23me3 is a heterochromatic mark in C. elegans” states that H3K23me1 is uniformly distributed (Figure 3—figure supplement 1) but it looks like H3K23me2 distribution across the genome is just as uniform and doesn't cluster at the constitutive heterochromatic regions as H3K23me3 or H3K9me3.These results and distributions all need quantifying.

In order to better address this point and make a clearer understanding we performed box plot analysis of the chromosome coverage and split each chromosome into three sections: the heterochromatic arms and the euchromatic center (Figure 4—figure supplement 2). We compared these regions for each modification in order to better assess the distribution of the marks. In this analysis, we found that your observations were correct: H3K23me2 has a much more uniform distribution than H3K9me3. This analysis also showed that there is a much weaker correlation between H3K23me3 and H3K9me3 chromosome three. By presenting this data in the updated manuscript, readers will get a more accurate view of the modifications’ distributions.

Results section: There is no reference to Figure 4B in the text (endogenous hrde-1 target).

This has been amended.

Results section: Figure 4 should be renamed Figure 4C-G as these represent the genome-wide data mentioned in the sentence.

This has been amended.

Results section: Figure 4—figure supplement 1 shows slightly stronger H3K9me3 reduction in regions of germline nuclear RNAi-mediated heterochromatin (GRH). However, regions of germline nuclear RNAi-mediated transcriptional silencing (GRTS) show even a stronger depletion of H3K9me3 in set-32 mutants compared to met-2 set-25 double mutants. This is not in line with the statement of the author "As in published works, we found that H3K9me3 has a larger requirement of MET-2 and SET-25 than SET-32" and the conclusion that this further supports different functions of SET-32 and MET-2/SET-25.

We apologize for the confusion. Upon revisiting our published works, we realized that most of the data presented in the original Figure S7 is available in (PMID: PMC5311726, Figure S3), these data have been removed from the manuscript. In the whole genome there are more *met-2 set-25*-dependent H3K9me3 loci than *set-32*-dependent H3K9me3 loci.

However, we see a similar dependence on *set-32* and *met-2 set-25* in the GRTS and GRH regions. There is more H3K9me3 loss (in GRH regions) in *set-32* than *met-2 set-25*, but we think such small difference is not biologically significant. We observed that *set-32* mutant has H3K9me3 defect and *met-2 set-25* mutant has H3K23me3 defect. The functional significance of the crosstalk is unknown and requires future investigation. We have updated the text to reflect these findings.

[Editors' note: further revisions were suggested prior to acceptance, as described below.]

Reviewer #2:The authors have addressed some major concerns and the paper could be accepted for publication.I, though, noticed that many ChIP experiments do not have biological replicates.This is especially relevant the second paragraph of subsection “set-32 and hrde-1 are required for nuclear RNAi-induced H3K23me3”; where the authors try to understand the loss of H3K23me3 in different mutant backgrounds.I suggest to remove from this paragraph the sentence " These findings indicate that the regulation of H3K23me3 at nuclear RNAi targets is complex, and likely to involve other HMTs.", as this cannot be concluded based on experiments lacking biological replicates and therefore not being quantitative. The authors should also mention in the figure legends the number of biological replicates for each experiment, each time, as otherwise for the reader it is hard to find this info.

We have addressed this concern in both the Materials and methods section, supplementary files and figure legends. For example, we made a new subsection “ChIP-seq replicates” listing the number of replicates for each experiment. For most experiments there are at least two biological replicates. For *oma-1* heritable RNAi there is only 1 replicate and for H3K23me2 and H3K23me1 there is only one replicate for each experiment. All the rest have two to three replicates.

Reviewer #3:Unfortunately the authors have not been able to definitively address the most important referee concerns – that the multi-generation 'inheritance' of H3K23me3 may be a secondary consequence of the transmission of small RNAs or other histone modifications and the lack of 'function' for H3K23me3. However, I do think this is an interesting body of work.For the reasons stated in the original reviews, I think the title has the potential to be misleading to a general audience: 'transgenerational heritable histone modification' is a strong statement and something that has not been directly demonstrated and I think this should be toned down.

We agree that at this point we do not know the mechanism by which dsRNA-induced H3K23me3 persists in the later generations. We have changed our language to reflect this. We no longer use the phrase “transgenerationally inherited”. In the revised manuscript, we replaced it with either “transgenerational epigenetic effect” or “transgenerational histone modification”.